# Histone Deacetylases and Their Isoform-Specific Inhibitors in Ischemic Stroke

**DOI:** 10.3390/biomedicines9101445

**Published:** 2021-10-11

**Authors:** Svetlana Demyanenko, Valentina Dzreyan, Svetlana Sharifulina

**Affiliations:** Laboratory of Molecular Neurobiology, Academy of Biology and Biotechnology, Southern Federal University, pr. Stachki 194/1, Rostov-on-Don 344090, Russia; dzreyan@sfedu.ru (V.D.); svetlana.sharifulina@gmail.com (S.S.)

**Keywords:** ischemic stroke, epigenetics, histone deacetylase, histone deacetylase inhibitor

## Abstract

Cerebral ischemia is the second leading cause of death in the world and multimodal stroke therapy is needed. The ischemic stroke generally reduces the gene expression due to suppression of acetylation of histones H3 and H4. Histone deacetylases inhibitors have been shown to be effective in protecting the brain from ischemic damage. Histone deacetylases inhibitors induce neurogenesis and angiogenesis in damaged brain areas promoting functional recovery after cerebral ischemia. However, the role of different histone deacetylases isoforms in the survival and death of brain cells after stroke is still controversial. This review aims to analyze the data on the neuroprotective activity of nonspecific and selective histone deacetylase inhibitors in ischemic stroke.

## 1. Ischemic Stroke Treatment Challenges

Stroke is one of the leading causes of death in the world [1]. About 5 million people die every year. No more than 20% of surviving patients can return to their previous job. 2/3 of strokes occur in people over 65. However, strokes are getting younger. In recent years, about 20% of cerebrovascular accidents have been reported in people aged fewer than 50, and this number is steadily increasing. Thus, the stroke problem is of extreme medical and social importance [2,3,4,5,6,7].

In ischemic stroke (about 80% of all strokes) the occlusion of cerebral arteries by a thrombus, atherosclerotic plaque, spasm, or abrupt changes in blood pressure sharply decreases or stops the blood supply of the brain tissue [4,5,6,8]. Cerebral ischemia may be the presenting manifestation of hematological diseases and hematological disorders account for about 1.3% of all causes of acute stroke [9,10].

Stroke is a developing over time multi-stage process. It starts from the primary minor changes and leads to the formation of a penumbra, death or restoration of its cells, and to the irreversible structural damage of brain tissue causing neurodegeneration. A huge number of pathophysiological and biochemical processes in intracellular and intercellular signaling, proteolysis, and regulation of the transcriptional activity of the genome are occurred in between these phases. Each of these stages is the time point of possible application of anti-stroke drugs. Many compounds tested in animal and cell models with the exception of tissue plasminogen activator (tPA) or endovascular thrombectomy [11] reduced apoptotic cell counts, increased infarction size, and improved neurological deficits after stroke [12,13,14]. However, none of these drugs have been successful in clinical trials. Probably, the complex pathophysiology of cerebral ischemia, the complexity of signaling cascades and differences in the mechanisms of the acute and restorative phases of stroke are the main reasons for failures in translating experimental studies into clinical practice.

Numerous studies searching targets for neuroprotection have highlighted the importance of multimodal stroke therapy. An example of such a strategy, which has been shown to be effective in various models of ischemia, is the inhibition of histone deacetylases (HDACs) [15,16,17,18,19]. This review analyzes data on the neuroprotective activity of nonspecific HDACs inhibitors (iHDACs) and selective iHDACs.

## 2. Histone Deacetylases 

Cerebral ischemia generally reduces the global level of gene expression due to suppression of acetylation of histones H3 and H4 [20,21]. The antibody microarray study and immunofluorescence microscopy have shown the twofold decrease in the acetylation of lysine 9 in histone H3 (H3K9Ac) in the ischemic penumbra at 1 h after photothrombotic stroke in the rat brain cortex, and more than fourfold decrease at 4 and 24 h. H3K9Ac was shown to localize exclusively in the neuronal, but not astroglial nuclei. These effects could be associated either with downregulation of histone acetyltransferases, or with overexpression of histone deacetylases [22]. Histone acetyltransferases (HATs) acetylate lysine residues in the histone tails. This promotes DNA unfolding and chromatin decondensation that facilitates transcription and protein synthesis. Histone deacetylases remove the acetyl groups from histones. This leads to formation of the transcriptionally inactive heterochromatin, in which gene expression is hindered. To maintain the transcriptionally active state of chromatin, HATs and HDACs should work together [23,24,25].

Among a large number of cellular non-histone proteins deacetylated by HDACs are transcription factors and co-regulators (e.g., c-MYC, HMG, YY1, EKLF, E2F1, factors GATA, HIF-1α, MyoD, NF-κB, and FoxB3), tumor suppressor proteins (e.g., p53, RUNX3), signaling mediators (e.g., STAT 1 and 3, β-catenin, and SMAD7), steroid receptors (e.g., androgens, estrogen, and SHP), and chaperone proteins and nuclear transport proteins (e.g., α-tubulin, importin-α, cortactin, Ku70, and HSP90) [26,27,28]. These proteins determine the growth, differentiation, migration, and activity of the protein determining cell survival both in normal conditions and under damage [27]. Thus, deacetylation-dependent signaling pathways play a crucial role in cell homeostasis.

Four classes of HDACs are distinguished in mammals according to their functions, intracellular localization, and expression patterns (Figure 1). Class I includes HDAC1, HDAC2, HDAC3, and HDAC8, class II consists of HDAC4, HDAC5, HDAC6, HDAC7, HDAC9, and HDAC10), and class IV contains only one HDAC11. All of them are zinc-dependent enzymes. Sirtuins use nicotinamide adenine dinucleotide NAD^+^ as a cofactor. They form the class III histone deacetylases. HDACs are evolutionary conservative [29,30,31,32]. 

## 3. Role of Histone Deacetylases in Cell Damage and Recovery after Cerebral Ischemia

### 3.1. Class I HDACs

The first class HDACs is widely represented in the brain [31]. HDAC1 localizes both in the neuronal nuclei, and in the cytoplasm, where it deacetylates some cytoplasmic proteins. HDAC2 localizes exclusively in the neuronal nuclei [22,33]. HDAC1 suppresses the production of proteins, which regulate the cell cycle in somatic cells. It also contributes to cell protection against DNA damage [34]. HDAC1 can serve as a molecular switch between neuronal survival and death [35]. HDAC2 regulates apoptosis in the ischemic penumbra [33,36]. HDAC1 and HDAC2 can be included in the multienzyme complexes Sin3, NuRD, CoREST, or NODE that suppress transcription of different sets of target genes [37,38,39]. The complex CoREST suppresses genes involved in synaptic plasticity and post-stroke recovery [40,41] (Figure 1).

The upregulation of HDAC2, but not HDAC1, in the PTS-induced penumbra was associated with development of apoptosis [22]. Other authors also showed the critical role of HDAC2 in death of neurons in the peri-infarction area after ischemic stroke. The upregulation of HDAC2 observed in the early recovery phase from five to seven post-stroke days reduced survival of neurons and augment neuroinflammation. HDAC2 targeting is apparently a novel therapeutic strategy for stroke recovery [36]. MCAO-induced overexpression of HDAC2 decreased the number of synapses, impaired synaptic plasticity, reduced memory, and deteriorated other cerebral functions [42,43]. Knockdown or knockout of the HDAC2 gene restored brain functions due to plasticity of the surviving neurons in the peri-infarction zone [43,44].

The overexpression of HDAC1 and HDAC2 in ischemic penumbra neurons and white matter glial cells was observed in the mouse brain during the early regeneration period, 1 week after MCAO. Their levels in the infarct core, oppositely, decreased [31]. Long-term overexpression of HDAC2 and HDAC8 was observed in neurons and astrocytes at 3‒14 days after photothrombotic stroke in the mouse cerebral cortex [45]. Thus, HDAC2 deteriorates synaptic processes, impairs memory, disturbs various cerebral functions, and stimulates apoptosis in the ischemic brain. 

HDAC3 also deacetylates histones H3 and H4 and some non-histone proteins. It also contributes to regulation of DNA replication and repair. The complex HDAC3/NCOR/SMRT is essential for maintaining chromatin structure and genome stability [46]. Overexpression of HDAC3, HDAC6, and HDAC11 was observed in the ischemic penumbra 3 and 24 h after MCAO, and persisted for a week after reperfusion. The upregulation of HDAC3 and HDAC6 in the mouse cortical neurons was also observed in vitro in the neuronal cell culture. The inhibition of HDAC3 or HDAC6 expression by the short hairpin shRNA increased cell viability. This suggested their involvement in ischemia-induced neurotoxicity [47]. The ischemia-induced neurotoxicity of HDAC3 was demonstrated in other studies [35,48,49]. The neurotoxic effect of HDAC3 was associated with its binding to HDAC1. Actually, the knockdown of HDAC3 suppressed the neurotoxicity of HDAC1, whereas HDAC1 knockdown suppressed the neurotoxicity of HDAC3. HDAC3 and HDAC1 interact with histone deacetylase-related protein (HDRP), a shortened form of HDAC9, whose expression was reduced during neuronal death. The interaction between HDRP and HDAC1, but not HDAC3 protected neurons. HDRP inhibited the HDAC1/HDAC3 interaction and prevented the neurotoxic effect of any of these proteins. This is a possible mechanism of HDAC1-mediated action as a switch “survival/death” in cerebral neurons. HDAC1 interaction with HDRP promotes neuron survival, whereas its interaction with HDAC3 leads to apoptosis [50]. On the other hand, HDAC3 was shown to suppress the production of the pro-apoptotic transcription factor E2F1 in neurons, and thus to contribute to survival of these cells [51].

Another class I histone deacetylase HDAC8 is present mainly in the cytoplasm of neurons and astrocytes of the cerebral cortex, amygdala, hippocampus, and hypothalamus [45,52]. The expression of HDAC8 in the mouse cortical neurons and astrocytes increased significantly during the recovery period, from 3 to 14 days after photothrombotic stroke [43].

### 3.2. Class II HDACs

The data on the role of HDAC4 in neurodegeneration and neuroprotection are contradictory. On one hand, some authors have reported the ability of HDAC4 to maintain neuronal survival [53,54,55]. However, other authors did not find a dependence of neuronal survival on HDAC4 expression [56,57]. In cultured neurons, HDAC4 rapidly translocates into the nucleus under glutamate release, or decreased K^+^ concentration in the medium. This stimulated cell death [58,59]. The administration of brain-derived neurotrophic factor (BDNF) prevented nuclear translocation of HDAC4 [60]. On the contrary, inhibition or loss of calmodulin-dependent kinase IV (CaMKIV) stimulated HDAC4 accumulation in the neuronal nuclei [59,61]. Nuclear HDAC4 was shown to promote neuronal apoptosis by suppressing the activity of prosurvival transcription factors MEF2 (myocyte enhancer factor 2) and CREB (cAMP response element-binding protein) [58]. Other authors have reported that HDAC4 translocation into the nuclei of neurons, but not astrocytes, did not cause apoptosis in the MCAO-induced ischemic penumbra. Moreover, the nuclear localization of HDAC4 promoted post-stroke brain recovery [62]. The HDAC4 level in the cytoplasmic, but not nuclear fraction of the rat brain cortex decreased at 24 h after photothrombotic stroke [33]. The downregulation of HDAC4 and its relocalization into the neuronal nuclei continued during the recovery period, 2 weeks after stroke [63,64]. In the neuronal nuclei HDAC4 deacetylates histones H3 and H4 and decreases the levels of some prosurvival proteins that finally lead to the neuronal death [58,59,64]. Since HDAC4 was assumed to be inactive against histones, these effects could be mediated by its interaction with other nuclear proteins. Actually, HDAC4 was shown to exhibit deacetylase activity after interacting with the co-repressor complex HDAC3/NCOR [65]. Further studies of HDAC4 interactions with different proteins are needed to understand its role in survival and death of cerebral cells after stroke.

HDAC5, another member of class II histone deacetylases, is involved in neuronal differentiation and axon regeneration in the injured sensory neurons [66,67]. The overexpression of HDAC5 and its nuclear localization was shown to be associated with apoptosis of the cultured neurons from the cerebellar granular layer [68]. After transient MCAO, HDAC5 suppressed the antiapoptotic effect of the transcription factor MRTF-A (myocardial transcription factor-A) in the rat brain neurons [69]. HDAC5 expression in the ischemic penumbra decreased 1, 2, and 14 days after MCAO [47,64]. The downregulation of HDAC5 in the mouse cerebral cortex was observed at 3 days after photothrombotic stroke. However, the number of the apoptotic HDAC5-positive cells did not change [63]. Possibly, the decrease in the level of HDAC5 in cortical neurons was associated with the regeneration processes [70]. 

HDAC6 belongs to the IIb class of histone deacetylases. It is involved in various cellular processes such as degradation of damaged proteins, cell migration, and intercellular interactions [71]. One of the cytoplasmic substrates of HDAC6 is α-tubulin. Its deacetylation induced destabilization of microtubules in the course of cytoskeleton reorganization and axonal growth during post-stroke regeneration [72]. In the mouse or rat brains HDAC6 presents not only in the cytoplasm, but also in the nuclei of some cortical neurons, but not astrocytes [33,63]. During the first two weeks after the photothrombotic stroke, HDAC6 was upregulated in the neurons not only in the penumbra, but also in the contralateral cerebral cortex, where it appeared in the neuronal nuclei. In the PTS-induced penumbra, HDAC6 co-localized with apoptotic neurons that indicated its involvement in neuronal apoptosis [57,63]. 

Thus, HDACs are widely represented in the brain. The expression of HDAC1, HDAC2, HDAC3, HDAC4, and HDAC6 increased in the ischemic penumbra. Some of them are located in the neuronal nuclei, some in the cytoplasm, and others—both in the nucleus and cytoplasm. Their functions after ischemic stroke differed. Some HDACs mediate prosurvival processes, whereas others are involved in neurotoxicity. HDAC2 and HDAC6 were apparently involved in apoptosis in the post-ischemic brain.

### 3.3. Sirtuins

Sirtuins (SIRT) are class III histone deacetylases. The coenzyme nicotinamide adenine dinucleotide (NAD^+^) makes sirtuins sensitive to metabolic and redox changes [73]. In mammals, seven sirtuins have been identified. Of these, SIRT1 and SIRT6 are localized mainly in the cell nuclei, SIRT7 in the nucleoli, SIRT2 in the cytoplasm, and SIRT3, SIRT4, and SIRT5 are mitochondrial proteins [74]. Sirtuins deacetylate a variety of substrates such as transcription factors, enzymes, and histones. They control diverse biological processes including metabolism, cell growth, aging apoptosis, and autophagy [75]. In the present review we focus on the role of non-mitochondrial SIRT1, SIRT2, and SIRT6 in the brain damage and recovery after ischemic stroke.

SIRT1 content in the brain is higher than in other organs [74]. In the hippocampus it regulates synaptic plasticity and memory. Since SIRT1 deacetylates histones and various transcription factors [72,76], and, also, has the chaperone-like activity [77], its subcellular localization is of significant importance for its functioning. The presence of the nuclear localization signal (NLS) and the nuclear export signal (NES) in the SIRT1 molecule allows it to shuttle from the nucleus to the cytoplasm and back that was assumed to be required for synaptic plasticity and memory formation [78,79]. The subcellular location of SIRT1 changed during brain development and in response to physiological and pathological stimuli [79,80]. 

SIRT1 mediates neuroprotection after ischemic stroke, traumatic brain injury, and neurodegenerative diseases. It regulates neurogenesis, neurite outgrowth, and gliogenesis, which are involved in postischemic brain regeneration [76,81]. In the SIRT1 knockout mice, MCAO induced greater cerebral infarction than in control animals [82]. On the contrary, mice overexpressing SIRT1 were more resistant to ischemia than control animals [83]. The activation of SIRT1 by resveratrol reduced the MCAO-induced infarction volume [84]. SIRT1 was overexpressed in the ischemic penumbra 7 days after MCAO in the mouse cerebral cortex [82]. Nuclear SIRT1 was reported to prevent apoptosis by deacetylation of proteins p53 [85], FOXO [86], and Ku70 [87]. On the contrary, SIRT1 localized in the cytoplasm enhanced caspase-dependent cell apoptosis [88]. Nevertheless, the translocation of SIRT1 into the cytoplasm was not associated with cell apoptosis in the peri-infarct area at 7 days after photothrombotic stroke in the mouse cerebral cortex. In this case, the cytoplasmic localization of SIRT1 was associated with the upregulation of synaptophysin and GAP-43 that mediate the axon outgrowth and restoration of synaptic connections [89]. The cytoplasmic SIRT1 was shown to enhance the neurite outgrowth that was induced by nerve growth factor NGF. Oppositely, inhibitors of SIRT1 or SIRT1-siRNA significantly reduced this effect [90]. 

SIRT2 is expressed predominantly in oligodendrocytes and in the myelin-rich regions of the ischemic brain. It was not found in astrocytes, microglia, or neurons [91]. However, other authors reported the presence of SIRT2 in the cytoplasm of neurons, but not astrocytes in the mouse cerebral cortex [89,92]. In the cytoplasm, SIRT2 such as HDAC6 regulates the microtubule dynamics through deacetylation of α-tubulin [93]. SIRT2 and HDAC6 can deacetylate α-tubulin either together [94], or separately [93]. Interestingly, the inhibition of SIRT2 increased acetylation of the microtubular α-tubulin mainly in the perinuclear zone, whereas inhibition of HDAC6 caused the general hyperacetylation of microtubules throughout the cell [95]. Although some studies pointed to the pathological role of SIRT2 after cerebral ischemia [76,96], the functions of SIRT2 in the ischemic brain are possibly more complicated than only pathological or only neuroprotective. Indeed, transient MCAO reduced the expression of SIRT2 and its translocation into the neuronal nuclei that played a neuroprotective role [96]. On the contrary, the overexpression of SIRT2 in the cytoplasm of the cerebellar neurons or in vitro in the differentiated PC12 cell line was shown to induce apoptosis [77,97]. Sirt2 was shown to mediate the myelin-dependent neuronal dysfunction during the early phase after MCAO in the mouse brain. Notably, the dynamics of Sirt2 mRNA and the protein level after ischemia differed [91]. 

SIRT6 was found in both: cerebral neurons and astrocytes [89,98]. It deacetylates mainly the lysine residues 9 and 56 in histone H3 that possibly represses genes associated with aging [99]. The role of SIRT6 in ischemia is still unclear and controversial. On one hand, SIRT6 protected the brain from postischemic reperfusion injury due to stimulation of transcription factor Nrf2 (nuclear factor-like (erythroid 2)-like 2), which regulates the expression of antioxidant proteins and suppresses oxidative stress [100,101]. SIRT6 was co-expressed with GAP-43, a marker of axon growth and synapse formation, at 14 days after photothrombotic stroke in the mouse cerebral cortex [89]. Whether SIRT6 functions as a part of the multiprotein complex in the postsynaptic membranes [102], or it regulates the neurite growth during the post-stroke recovery phase should be further studied. SIRT6 immunofluorescence was not observed in apoptotic cells in the PTS-induced penumbra [89]. On the other hand, its overexpression in cultured neurons under oxygen and glucose deprivation was associated with necrosis of cortical cells [103]. 

Thus, sirtuins, SIRT1 and SIRT6, are involved in the postischemic brain regeneration. SIRT1 regulates synaptic plasticity, memory, neuritogenesis, neurogenesis, and gliogenesis. SIRT6 protects neurons and astrocytes from the post ischemic reperfusion injury via stimulation of the transcription factor that regulates the production of antioxidant proteins Nrf2. 

The HDAC inhibitors that have been developed to date are capable of inhibiting almost all HDAC isoforms of these four classes with varying degrees of specificity.

## 4. Pan-Inhibitors of Histone Deacetylases in Cerebral Ischemia

Inhibitors of different histone deacetylases that efficiently protect the brain from ischemic injury belong to two chemical groups: (a) Small carboxylates: valproic acid (VPA), sodium butyrate (SB), and sodium 4-phenylbutyrate (4-PBA), and (b) Hydroxamate-containing compounds: suberoylanilide hydroxamic acid (SAHA, Vorinostat), trichostatin A (TSA), and others. They protect the brain against excitotoxicity, oxidative stress, ER stress, apoptosis, inflammation, and BBB breakdown. They also induce angiogenesis, neurogenesis, and migration of stem cells to the damaged brain regions that improves functional recovery after cerebral ischemia. HDAC inhibitors are the promising neuroprotectors for treating ischemic stroke [15,16,17,18].

VPA, a pan-HDAC inhibitor, was shown to reduce brain injury in various stroke models. It improved the functional outcome, and demonstrated the anti-inflammatory activity [104,105,106]. VPA induces different proteins such as NeuroD, Math1, Ngn1, and p15, which contribute to differentiation of neural precursors in the hippocampus [107]. VPA administration during 7 days after MCAO considerably improved the neurological outcome in rats. This effect was associated with enhanced white matter repair and neurogenesis [108,109]. The VPA treatment increased survival of oligodendrocytes and caused the generation of new oligodendrocytes. These effects were associated with the increased density of myelinated axons in the ischemic boundary around the infarction core. At the molecular level, VPA increased the acetylation of histone H4 and caused overexpression of glutamate transporter 1 (GLT1) in neuroblasts. It also increased the number of new neurons [109]. Prolonged application of VPA during two weeks after MCAO increased the acetylation of histones H3 and H4 in the rat brain, and caused the upregulation of transcription factor HIF-1α (hypoxia-inducible factor-1α) and its downstream pro-angiogenic molecules such as vascular endothelial growth factor (VEGF) and matrix metalloproteinases MMP2 and MMP9. This enhanced the microvessel density and promoted functional recovery [106]. VPA suppressed the nuclear translocation of the NF-κB subunit p65, reduced activity of matrix metalloproteinase MMP9, and restored the BBB integrity that was broken after stroke [110]. 

Other pan-HDAC inhibitors SB and 4-PBA showed similar neuroprotective activity [111,112,113]. SB stimulated proliferation of neuronal progenitor cells and neurogenesis in the SVZ and DG zones of the rat brain after MCAO. It increased the levels of acetylated histone H3, neural cell adhesion molecule nestin, glial fibrillary acidic protein, transcription factor CREB (phospho-cAMP response element-binding protein), and brain-derived neurotrophic factor (BDNF) that were reduced after cerebral ischemia [111]. The neuroprotective effect of SB was also associated with inhibition of oxidative stress, reduction of BBB permeability, and anti-inflammatory action [110]. Notably, VPA and SB stimulated neurogenesis in the peri-infarct regions during the post-stroke recovery period [108,109,111]. SB effect on proliferation, differentiation, and migration of the neural precursor cells was mediated by the BDNF-TrkB signaling pathway [114]. Microglia-mediated neuroinflammation is an important component of the stroke-induced brain pathology. As shown in vitro, SB reduced the acetylation of histone H3 that was increased in the activated microglial cells. It also altered the transcription of pro-inflammatory genes Tnf-α, Nos2, Stat1, and Il6. Simultaneously, SB induced the expression of anti-inflammatory genes regulated by the IL10/STAT3 pathway. In the microglia of mice subjected to MCAO, SB reduced the expression of pro-inflammatory proteins TNF-α and NOS2, and stimulated expression of the anti-inflammatory mediator IL10. Therefore, HDAC inhibition by SB turned the microglial activity in the ischemic brain from the pro-inflammatory to the anti-inflammatory mechanism [115].

The application of SAHA protected the rat brain from MCAO-induced ischemia [116]. It prevented the ischemia-induced decrease of the histone H3 acetylation level and reduced the infarct volume. It also increased the levels of neuroprotective proteins Hsp70 and Bcl-2 in both control and ischemic brains [117]. The intraperitoneal administration of SAHA at 12 h after transient MCAO reduced the infarction volume in the mouse brain and improved the post-stroke outcome. It reduced the level of pro-inflammatory cytokines and inhibited the microglia-mediated inflammatory response [118].

TSA increased survival of cultured neuronal cells after oxygen and glucose deprivation. It also decreased the stroke-induced infarct volume in the mice brain. These effects were associated with the activation of antioxidant processes. TSA mediated HDAC inhibition and activated the transcription factor Nrf2, which regulates expression of diverse antioxidant proteins. As a result, heme oxygenase 1, NAD(P)H:quinone oxidoreductase 1, and glutamate-cysteine ligase were overexpressed and mediated neuroprotection in the neuronal culture and in the ischemic brain [106,116].

These HDAC inhibitors are non-selective. The neuroprotective effects of non-selective inhibitors such as TSA, SB, and SAHA have been well reviewed [119,120,121,122]. They inhibit a group of HDACs that belong to class I, or II, or both. Some HDACs in these groups are involved in ischemia-induced cell death, whereas others participate in the neuroprotective processes. It is of interest to use more selective inhibitors in order to affect only the pathogenic HDACs in the ischemic brain. 

## 5. Selective Inhibitors of Histone Deacetylases in Cerebral Ischemia

### 5.1. Inhibition of Class I HDACs

In the work of Lin and colleagues photothrombotic stroke impaired neuronal survival and neuroplasticity in the mouse brain, deteriorated motor functions, and stimulated neuroinflammation [36]. These effects were associated with the upregulation of HDAC2 in the peri-infarct zone from 5 to 7 days after PTS. Since the absolute selective inhibitors of some HDACs are not available, the author used several HDACs inhibitors with the limited anti-HDAC selectivity: MGCD0103 that inhibits HDAC1, HDAC2, and HDAC3; SAHA that inhibits mostly HDAC1 and HDAC2, and TMP269, a class II inhibitor that acts on HDAC4, HDAC5, HDAC7, and HDAC9. The comparison of their effects showed the critical role of HDAC2 inhibition in the restoration of brain function, whereas HDAC2 overexpression exacerbated the stroke-induced functional disorders. HDAC2 inhibition at 5 to 7 post-stroke days promoted survival and neuroplasticity of neurons, suppressed neuroinflammation, recovered motor functions, and improved the stroke outcome. The inhibition of other HDAC isoforms was ineffective [36].

The recent inhibitory analysis also confirmed the participation of HDAC2 in the PTS-induced injury of the mouse cerebral cortex. The administration of α-phenyltroplon that inhibits both HDAC2 and HDAC8 [123], or MI-192, an inhibitor of HDAC2 and HDAC3 [124], suppressed apoptosis of cortical cells in the penumbra and decreased the infarct volume after photothrombotic stroke in mice. These effects contributed to the restoration of cerebral functions. However, the selective HDAC3 inhibitor BRD3308 was ineffective. Hence, HDAC2 was involved in PTS-induced apoptosis in the mouse brain, and its inhibition was beneficial against ischemic stroke [57,123] (Table 1).

It has been shown that in ischemia reperfusion kidney injury in HDAC1 or HDAC2 knockout mice, deletion of HDAC2 but not HDAC1 leads to an ischemic damage decrease [134].

Recently, it was shown that in ischemic reperfusion kidney injury in mice with inducible HDAC1 or HDAC2 knockout, it is the deletion of HDAC2, but not HDAC1, that leads to a decrease in ischemic damage.

Currently, new carbamide based inhibitors of ortho-amino anilides have been developed which show high selectivity for HDAC2 compared to the highly homologous isoform HDAC1. These kinetically selective HDAC2 inhibitors (BRD6688 and BRD4884) increased the acetylation of histones H4K12 and H3K9 in primary mouse hippocampal cell cultures and improved learning and memory in a model of neurodegenerative disease in mice [135].

The selective HDAC3 inhibitor RGFP966 reduced infarction size and alleviated neurological deficits after MCAO by reducing the inflammatory response. RGFP966 enhances STAT1 acetylation and subsequently attenuates STAT1 phosphorylation that may lead to AIM2 inflammasome downregulation by RGFP966 [136]. Inhibition of HDAC3 by RGFP966 was protective against I / R cerebral injury in in vivo and in vitro models of diabetes by modulating oxidative stress, apoptosis, and autophagy that may be mediated by upregulation of Bmal1 [137].

RGFP966 administration mimicked the neuroprotective effect of ischemic tolerance after MCAO causing a decrease in infarction volume and neurological deficits. The mechanism of the protective action of the HDAC3 inhibitor may be based on the weakened recruitment of HDAC3 to the promoter regions after preconditioning potentiating the initiation of transcription of genes including Hspa1a, Bcl2l1, and Prdx2 [138].

The selective HDAC8 inhibitor PCI-34051 has a neuroprotective effect. However, it was found that HDAC8 is not involved in the PCI-34051 mechanism of action [139]. Using BRD3811, an inactive PCI-34051 ortholog, has been shown to exhibit strong neuroprotective properties despite its inability to inhibit HDAC8. Research by Sleiman et al. (2014) [139] showed that the protective effects of small molecules containing hydroxamic acid, such as PCI-34051, are probably not related to direct epigenetic regulation through inhibition of HDAC8, but rather their neuroprotective effect is based on their ability to bind metals exhibiting antioxidant properties.

### 5.2. Inhibition of Class II HDACs 

Selective inhibition of the class IIa HDACs by MC1568 worsened brain recovery. This was associated with the inactivation of CREB and c-Fos and exacerbated neurological deficit [140]. However, recent studies have shown that pharmacological inhibition of class IIa HDAC by MC1568 reduces infarction volume and neurological deficit in rats after tMCAO [138]. The effect was associated with the activation of transcription of the ncx3 gene encoding the plasma membrane Naþ / Ca2þ exchanger 3 (NCX3), which plays a neuroprotective role in stroke.

LMK235, a selective inhibitor of HDAC, did not influence the infarct volume, apoptosis of penumbra cells, and the neurological deficit. This indicated that HDAC4 did not participate in the apoptosis of penumbral cells after photothrombotic stroke [57]. On the other hand, LMK235 dose-dependently inhibited MKK7 transcription and JNK / c-Jun activity, which protected cultured cerebellar granule (CGN) neurons from apoptosis caused by potassium deficiency [141].

HDAC6 inhibitors were reported to reduce ischemia-induced apoptosis in the rat brain [126,142], the retina [143], or the myocardium [144]. They inhibited oxidative stress by decreasing the acetylation of peroxiredoxin 1 [144]. The selective HDAC6 inhibitors tubastatin A or HPOB promoted post-stroke regeneration of the mouse brain. They reduced apoptosis of the penumbral cells, and decreased the infarction volume after photothrombotic stroke. These compounds also stimulated the axon outgrowth, as was evidenced by the overexpression of GAP43 and restoration of acetylation of α-tubulin [57,63]. Tubastatin A also recovered the acetylated state of fibroblast growth factor-21 (FGF-21) that contributed to restoration of brain functions [126]. 

Another selective HDAC6 inhibitor tubacin increases eNOS expression in vivo improving endothelial function in diabetic db / db mice and significantly reduces ischemic brain damage in a mouse stroke model [127].

Currently, new low molecular weight inhibitors of HDAC6 pyrimidine hydroxyl amide have been developed that are bioavailable in the brain when administered systemically, such as ACY-738 and ACY-775, which have antidepressant effects [145] and are effective in peripheral neuropathy [146]. Their effectiveness in cerebral ischemia has not been studied yet.

These data showed the involvement of different HDACs in some neuropathological processes. In particular, HDAC2 and HDAC6 are involved in the stroke-induced death of neurons in the ischemic brain. Their activity may be associated with deacetylation of histones H3 and H4, following chromatin condensation, and inhibition of protein biosynthesis in the cell. Additionally, some HDACs deacetylate cytoplasmic proteins and modulate their functions. HDAC inhibitors prevent these effects and improve cell viability. Notably, some HDAC inhibitors showed the efficient neuroprotection not only during 3‒6 h, but later, up to 5-7 days after ischemic stroke. Such delay may expand the therapeutic window that is very important for practical anti-stroke medicine.

### 5.3. Sirtuins Activators and Inhibitors 

Since the modulation of the sirtuin activity can have a beneficial effect on many diseases, there is growing interest in the development and testing of sirtuin activators and inhibitors [147,148]. Here we consider those compounds that activate or inhibit sirtuins under conditions of cerebral ischemia, discuss the data confirming their effectiveness in cellular and animal models of ischemia.

Almost all sirtuin activators are described only for SIRT1 (Table. 1) The most famous of the SIRT1 activators is resveratrol (3, 5, 4′-trihydroxy-trans-stilbene). It is a natural compound that activates SIRT1 and may help treat or prevent obesity, reduce carcinogenesis, and age-related decline in heart function and neuronal loss. Pharmacological modulation of SIRT1 can have a pronounced effect on the outcome of ischemic stroke [76].

Resveratrol applied to mice before and after reperfusion reduced neurological deficit and infarction volume in the MCAO model by increasing the expression of angiogenic factors MMP-2 and VEGF and the number of microvessels [129]. Preliminary administration of resveratrol to mice for 7 days reduced the expression and activity of MMP-9, increasing the viability of neurons [79]. Recent studies have shown that pre-intraperitoneally administered resveratrol reduced neurological deficits and infarction volume 24 hours after MCAO in male rats (30 mg/kg) [130]. Moreover, the same study showed that resveratrol attenuated neuronal cell death by increasing the phosphorylation level of Akt and GSK-3β [130].

A recent review by Ghazavi H et al (2020) [149] collected evidence that resveratrol may not only affect neuronal function, but also plays an important role in reducing neurotoxicity by altering glial activity by modulating a number of signaling pathways. Numerous studies indicate that resveratrol enhances anti-inflammatory effects and decreases inflammatory cytokines by acting on signaling pathways in microglia such as AMP-activated protein kinase (5′-adenosine monophosphate-activated protein kinase, AMPK), SIRT1, and SOCS1 (suppressor of cytokine signaling Resveratrol increases AMPK activity and inhibits GSK-3β (glycogen synthase kinase 3 beta) activity in astrocytes), which makes ATP available to neurons and reduces reactive oxygen species (ROS). In addition, resveratrol promotes microglial activation and increases survival oligodendrocytes, which can lead to the maintenance of post-stroke brain homeostasis [149].

Intraperitoneal combined administration of the MS-275 histone deacetylase I inhibitor (20 μg / kg) with resveratrol (680 μg / kg) to mice after tMCAO reduced infarction volume and neurological deficits, assessed 48 hours after tMCAO [150]. Then, 24 hours after administration of the drugs, a decrease in the binding of RelA to the Nos2 promoter was observed, which reduced the overexpression of a number of proteins associated with the activation of microglia and macrophages, which ultimately led to a decrease in leukocyte infiltration in the ischemic region [150].

In addition to sirtuins, resveratrol affects many other targets (e.g., kinases and ATP synthase) [151], so the molecular mechanisms found in physiological studies with resveratrol should be interpreted with caution. In addition, resveratrol can also inhibit SIRT1, SIRT3, or SIRT5 depending on the substrate used [152,153].

It should be noted that natural resveratrol suffers from low bioavailability and activity and, like early synthetic SIRT1 activators such as SRT1720, has low selectivity [147]. However, SRT1720 induced mitochondrial biogenesis, increasing the mitochondrial respiration rate and ATP level, reducing ischemia-reperfusion kidney damage [154,155]. SRT1720 accelerated the recovery of mitochondrial function after acute oxidative damage to renal proximal tubule cells [156]. The SRT1720 activator was also effective in ischemia-reperfusion myocardial injury [157]. In a model of brain oxidative stress in vivo, it was shown that SRT1720 has a neuroprotective effect, reducing neurological deficit by inhibiting poly (ADP-ribose) polymerase [158].

In addition to SRT1720, molecules structurally unrelated to resveratrol, such as SRT2104, SRT2379, and SRT3657, have been developed to stimulate the activity of sirtuins, but similar to SRT1720 these SIRT1 activators have not been shown to have a neuroprotective effect in models of ischemic stroke.

Other SIRT1 activators such as tetrahydroxystilbene glucoside (TSG) reduced the effects of OGD and MCAO in cultured neurons and animals, respectively [23]. Icariin is able to increase the expression of SIRT1 and its downstream target PGC-1α, which stimulates mitochondrial activity, both in the MCAO mouse model and in neurons treated with OGD. Incarion improved neurological parameters after MCAO in mice, reduced the size of myocardial infarction and cerebral edema [132]. The neuroprotective effect of incarion was reversed by inhibition of SIRT1.

Treatment of mice with Activator III reduced the infarction volume, while the SIRT1 and SIRT2 inhibitor sirtinol increased ischemic damage both in the model of ischemic stroke [82] and in the model of hemorrhagic stroke [159]. Sirtinol increased acetylation of p53 and nuclear factor κB (p65), which led to the activation of neuronal apoptosis [82,159].

Another SIRT1 activator, citicoline (CDP-choline) was effective for patients with moderate stroke [160]. Treatment with citicoline increased SIRT1 levels in rat brains after tMCAO concurrently with neuroprotection. Sirtinol blocked the decrease in infarction volume caused by citicoline, while resveratrol caused a strong synergistic neuroprotective effect with citicoline [160].

Thus, most studies provide evidence of a neuroprotective effect of SIRT1 activators. Inhibition of neuroprotective SIRT1 usually worsens stroke outcome [76]. On the other hand, the SIRT1 inhibitor nicotinamide can also protect neurons from excitotoxicity and cerebral ischemia [109]. Clinically used selective inhibitor SIRT1 Selisistat, also known as EX-527, reduces the volume of ischemic brain infarction and improves survival, but does not reduce neurological deficits associated with stroke [128]. In addition, the administration of EX-527 effectively increased the expression of metabolic enzymes associated with the regulation of necroptosis [128].

Besides SIRT1, modulation of the activity of other sirtuins can also influence the progress or outcome of cerebral ischemia.

The administration of AGK2, a potent inhibitor of SIRT2, had a neuroprotective effect in the MCAO model; neurological indicators were better compared to the control group [96]. Inhibition of SIRT2 by AGK2 or SIRT2 knockdown reduced cell death caused by hydrogen peroxide by decreasing ROS production in cell culture [97].

However, the use of another SIRT2 inhibitor AK-7 in animal stroke models did not show a neuroprotective effect in the MCAO mouse model [161]. Whereas the administration of other SIRT2 inhibitors AK-1 and AGK2 reduced the infarction volume by suppressing the proapoptotic signaling pathways AKT / FOXO3a and MAPK [162].

Thus, SIRT2 also performs contradictory functions in stroke [163]. Its effect probably depends on the localization of the enzyme [63] as well as on the stage of ischemic stroke. It is known that Sirt2 is able to mediate myelin-dependent neuronal dysfunction at an early stage after ischemic stroke [91].

Modulators for other sirtuins are less studied. To date, little is known about the effectiveness of the pharmacological modulators SIRT3-7 in cerebral ischemia.

## 6. Conclusions

The brain responses to ischemic stroke are very complex and dynamic. They involve a plethora of molecular processes that occur in different cerebral elements—neurons, glial cells, and blood vessels. The neuroprotective strategies in vivo have failed in human trials [164]. However, the previous presence of a transient ischemic attack is associated—in humans—with a good early outcome in nonlacunar ischemic strokes, thus suggesting a neuroprotective effect of transient ischemic attack possibly by inducing a phenomenon of ischemic tolerance [165]. After numerous unsuccessful searches of effective neuroprotective anti-stroke agents, which used the compounds affecting only one side of this complex process, such as glutamate-mediated excitotoxicity, calcium homeostasis, or different apoptosis stages, it became that the complex approach aimed at several targets in the ischemic cerebral tissue is needed. In addition to brain neuroprotection, it is also important to study and stimulate the processes involved in neurorepair in the adult brain, either from angiogenesis, neurogenesis, or synaptic plasticity, including through endogenous neurorepair phenomena [166]. 

The inhibitors of different histone deacetylases efficiently protected the animal brains from ischemic injury. They induce angiogenesis, neurogenesis, stem cell migration to the damaged brain regions in order to promote functional recovery after cerebral ischemia. Hence, HDAC inhibitors may be considered as the hopeful neuroprotector agents for treatment of ischemic stroke. The first generation HDAC inhibitors are non-selective. They inhibit not only the HDACs involved in the ischemia-induced cell death, but also those participating in the neuroprotective processes. Possibly, more selective inhibitors that affect specific pro-apoptotic HDACs in the ischemic brain may be promising neuroprotectors for treating ischemic stroke.

## Figures and Tables

**Figure 1 biomedicines-09-01445-f001:**
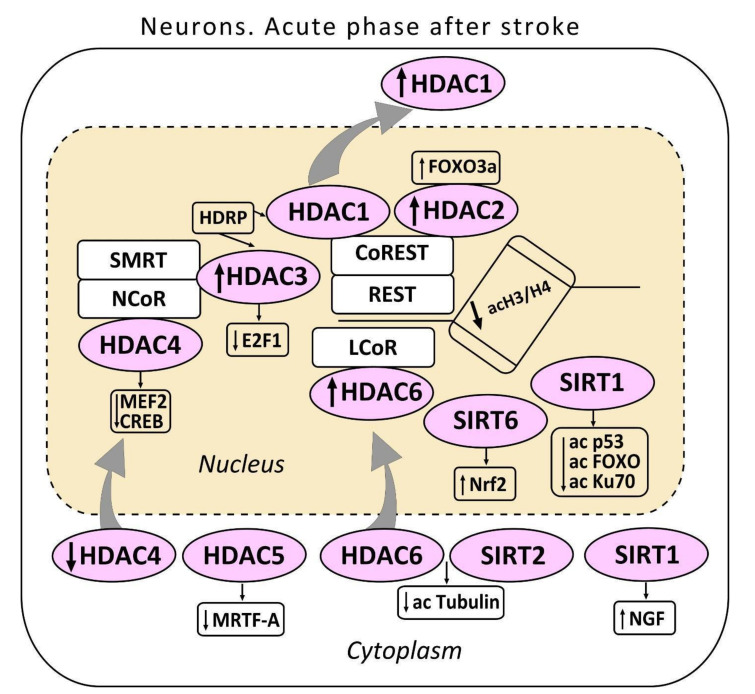
The acute phase after stroke in the neuron. The localization of histone deacetylases (HDAC 1-6) including sirtuins (SIRT 1,2) in the cell is shown. HDAC3 and HDAC1 interact with histone deacetylase-related protein (HDRP), a shortened form of HDAC9. HDRP inhibited the HDAC1/HDAC3 interaction. HDAC1 and HDAC2 are included in the repressor element-1 silencing transcription factor (REST) corepressor 1 (CoREST) complex. HDAC3 in the nuclear receptor co-repressor (NCOR)/silencing mediator for retinoid or thyroid-hormone receptors (SMRT) complex suppresses the production of the pro-apoptotic transcription factor E2F1. HDAC4 suppresses the activity of myocyte enhancer factor 2 (MEF2) and cAMP response element-binding protein (CREB). HDAC4 interacts with the co-repressor complex HDAC3/NCOR. HDAC5 suppressed myocardial transcription factor-A (MRTF-A). One of the cytoplasmic substrates of HDAC6 - α-tubulin is shown. SIRT1 deacetylates p53, FOXO, and Ku70 proteins. The cytoplasmic SIRT1 induces nerve growth factor NGF. SIRT6 stimulates the nuclear factor erythroid 2–related factor (Nrf2).

**Table 1 biomedicines-09-01445-t001:** Effects of selective HDAC inhibitors and activators in animal models of ischemic stroke.

Class I	Inhibitor	Model of stroke	Treatment time	Effective Doses	Effects on Stroke	Citations
HDAC1/2/3	MS-275, entinostat	Mouse, MCAO Mouse, MCAO	Post at 7 h Post at 0h, 24 h and 48 h	20μg/kg 200μg/kg 30 mg/kg	↓Infarct volume ↓Neurological deficit ↑Cell survival	[20] [125]
HDAC2/3	MI192	Mouse, PTS	Post for 3 days	40 mg/kg	↓Infarct volume ↓Apoptosis ↑Acetylated α-tubulin ↑GAP43 ↓Neurological deficit	[124]
HDAC2/8	α-phenyl tropolone	Mouse, PTS	Post for 7 days	10 mg/kg	↓Infarct volume ↓Apoptosis	[123]
HDAC6	Tubastatin A	Rats, MCAO Mouse, PTS	Post for 1 or 3 days Post for 4 or 7 days	25 mg/kg 40 mg/kg 25 mg/kg	↓Infarct volume ↓Apoptosis ↑Acetylated α-tubulin ↓Neurological deficit ↑FCF-21 ↑GAP43	[126] [22]
	Tubacin	Mouse, MCAO	Pre for 3 h	5 mg/kg	↓Infarct volume ↑eNOS	[127]
	HPOB	Mouse, PTS	Post for 7 days	10 mg/kg	↓Infarct volume ↓Apoptosis	[89]
SIRT1	EX-527	Rats, MCAO	Post at 6 h, 12 h or 24 h	10 μg	↓Infarct volume ↓Necroptosis ↑Survival	[128]
SIRT1 and SIRT2	Sirtinol	Mouse, MCAO	Post at 48h	10 mg/kg	↑Infarct volume ↑Apoptosis ↑Acetylated p53 and p65	[82]
SIRT2	AGK2	Mouse, MCAO	Post at 24 h	1 mg/kg	↓Infarct volume ↓Apoptosis ↓Neurological deficit ↓JNK, c-jun ↓AKT/FOXO3a	[128]
	Activators					
SIRT1	Resveratrol	Mouse, MCAO Mouse, MCAO Rats, MCAO	Post for 7 days Post for 7 days Post at 24 h	50 mg/kg 6800μg/kg 30 mg/kg	↓Infarct volume ↓Neurological deficit ↑MMP-2 ↑VEGF ↓Infarct volume ↓Neurological deficit ↓Infarct volume ↓Neurological deficit ↑pAkt ↑pGSK-3β	[129] [20] [130]
	Tetrahydroxystilbene glucoside (TSG)	Mouse, MCAO	Post at 24h	15 mg/kg 40 mg/kg	↓Infarct volume ↓Apoptosis ↓ROS	[131]
	Icariin	Mouse, MCAO	Post for 1, 3 or 7 days	100 mg/kg 200 mg/kg	↓Infarct volume ↓Brain edema ↓Neurological deficit ↑PGC-1α	[132]
	Activator III	Mouse, MCAO	Post at 48 h	10 mg/kg	↓Infarct volume	[82]
	Alpha-lipoic acid (ALA)	Mouse, MCAO	Post at 24 h	50mg/kg	↓Infarct volume Brain edema ↓Neurological deficit	[133]

Model of transient middle cerebral artery occlusion (MCAO), photothrombotic stroke (PTS).

## Data Availability

Not applicable.

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
