# Peer review of "Histone Deacetylases and Their Isoform-Specific Inhibitors in Ischemic Stroke"

_biomedicines, 2021, doi:10.3390/biomedicines9101445_

Round 1

Reviewer 1 Report

The authors present a well-conducted and scientifically interesting review on Histone deacetylases (HDACs) and their isoform-specific inhibitors in experimental ischemic stroke. The authors concluded that the inhibitors of different histone deacetylases efficiently protected the animal brains from ischemic injury. They induced angiogenesis, neurogenesis, stem cell migration to damaged brain regions to promote functional recovery after cerebral ischemia. However, possibly more selective inhibitors affecting specific pro-apoptotic HDACs in the ischemic brain may be promising neuroprotectors to treat ischemic stroke. The study is potentially interesting, but can be improved if the following minor considerations are addressed:  

1.Cerebral ischemia may also be the presenting manifestation of hematological diseases (Eur Neurol 1997;37:207-211). This is a noteworthy aspect that should also be emphasized in the Introduction (see and add this reference: Expert Rev Hematol 2016; 9: 891-901).  

2. It would be interesting to mention in the text the review by Krupinski et al (Curr Cardiol Rev 2010; 6: 238-244) in which the authors point out that in addition to brain neuroprotection it is also important to study and stimulate the processes involved in neurorepair in the adult brain, either from angiogenesis, neurogenesis or synaptic plasticity, including through endogenous neurorepair phenomena.  

3.It would be interesting to include in the text a comment regarding the fact that numerous neuroprotective strategies “in vivo” have failed in human trials (ex: citicoline, -Lancet 2012; 380: 349-357-). However, it would be helpful to mention that the previous presence of a TIA is associated -in humans- with a good early outcome in nonlacunar ischemic strokes, thus suggesting a neuroprotective effect of TIA possibly by inducing a phenomenon of ischemic tolerance (see and add this reference Cerebrovasc Dis 2004; 18. 304-311). Did the authors consider this aspect in their systematic review?    

Author Response

Thank you so much for your useful comments and suggestions. Please see the attachment with authors responses to the comments. 

Reviewer 2 Report

The present manuscript from Demyanenco et al. is a well-conducted review about HCADCs inhibitors that have some therapeutic potential in ischemic stroke. The authors analyze carefully all the different classes of HDACs, their inhibitors, and activators, and as result, they show the role of each HDAC as a potential target to develop new therapies against ischemic stroke. Although they do not cover all the current molecules described in the literature as potential inhibitors or activators of HDACs and Sirtuins, however, they give a nice description of those activities of the most important ones in all the different in vitro and in vivo models.

Author Response

Thank you for your time and comments. Please see the attachment with authors'  responses.
